

# Wearing surgical face mask has no significant impact on auscultation assessment

Ivana Folnožić[1], Marija Gomerčić Palčić[1,2], Matilda Sabljak[2], Ena Vučak[1], Luka Vrbanić[1], Marija Mandić Perić[1], Fanika Mrsić[3], Aljoša Šikić[4] and Ivan Ivanovski[5]

[1] Division of Pulmonology, Department of Internal Medicine, Sestre Milosrdnice University Hospital Center, Zagreb, Croatia
[2] School of Medicine, University of Zagreb, Zagreb, Croatia
[3] Division of Clinical Immunology and Rheumatology, Department of Internal Medicine, Sestre Milosrdnice University Hospital Center, Zagreb, Croatia
[4] Department of Emergency Medicine, Sestre Milosrdnice University Hospital Center, Zagreb, Croatia
[5] Department of Anesthesiology, Sestre Milosrdnice University Hospital Center, Zagreb, Croatia

## ABSTRACT

**Objective**. During the COVID-19 pandemic, universal mask-wearing became one of the main public health interventions. Because of this, most physical examinations, including lung auscultation, were done while patients were wearing surgical face masks. The aim of this study was to investigate whether mask wearing has an impact on pulmonologist assessment during auscultation of the lungs.

**Methods**. This was a repeated measures crossover design study. Three pulmonologists were instructed to auscultate patients with previously verified prolonged expiration, wheezing, or crackles while patients were wearing or not wearing masks (physician and patients were separated by an opaque barrier). As a measure of pulmonologists' agreement in the assessment of lung sounds, we used Fleiss kappa (K).

**Results**. There was no significant difference in agreement on physician assessment of lung sounds in all three categories (normal lung sound, duration of expiration, and adventitious lung sound) whether the patient was wearing a mask or not, but there were significant differences among pulmonologists when it came to agreement of lung sound assessment.

**Conclusion**. Clinicians and health professionals are safer from respiratory infections when they are wearing masks, and patients should be encouraged to wear masks because our research proved no significant difference in agreement on pulmonologists' assessment of auscultated lung sounds whether or not patients wore masks.

## INTRODUCTION

Since its outbreak in December 2019, severe acute respiratory syndrome coronavirus 2 (SARS-CoV-2) that causes novel coronavirus disease 2019 (COVID-19) has caused 7,010,568 deaths worldwide (*World Health Organization, 2024*; https://data.who.int/dashboards/covid19/deaths?n=c, 14th January 2024). Since the beginning of the pandemic,

Corresponding author
Marija Gomerčić Palčić,
marijagomercic@yahoo.com

many government health care agencies have recommended community-wide use of face masks as a low-cost and available epidemiologic tool for decreasing viral transmission, especially in patients with chronic diseases (*European Centre for Disease Prevention and Control , 2021a*; *European Centre for Disease Prevention and Control , 2021b*). Several studies have demonstrated that wearing face masks leads to a reduction in virus transmission (*Seto et al., 2003*; *Jefferson et al., 2011*; *Smith et al., 2016*; *Chu et al., 2020*) although more randomized control studies are needed to confirm these findings. Lung sound auscultation is still one of the key parts of a physical examination that is helpful in identifying respiratory pathology even when chest radiography findings appear normal (such as detecting crackles in patients with interstitial lung disease or detecting wheezes in bronchoobstruction). Other advantages of auscultation also lie in its widespread availability and affordability (*Fajardo & Davis, 2022*). According to the European Respiratory Society (ERS) Task Force on Respiratory Sounds, respiratory sounds should be divided into lung sounds and other (*e.g.*, pleural rub, grunting, and snoring). Lung sounds should then be divided into normal (basic) sounds and adventitious sounds (*Pasterkamp et al., 2016*).

Since the beginning of the COVID-19 pandemic, most physical examinations have been done while patients were wearing surgical face masks, so the aim of our study was to investigate whether epidemiological recommendations of wearing surgical face masks have an impact on pulmonologist assessment during auscultation, as well as the reliability of performance under this condition.

## MATERIALS & METHODS

This was a repeated measures crossover design study. We included 50 patients between November and December 2022 who were being treated at the Division of Pulmonology, Department of Internal Medicine, Sestre Milosrdnice University Hospital Center. All patients were older than 18 years and both genders were represented. Only patients with previously verified pathological lung auscultation finding (prolonged expiration, wheezing, or crackles) were included. All patients signed informed consent before being included in the study. The study was approved by the University Hospital Center Sestre Milosrdnice Ethical Committee (approval 251-29-11-21-03). Three pulmonologists were instructed to auscultate patients in the outpatient clinic (part of the forementioned division) and physicians did not sign informed consent. Inclusion criteria were stable chronic lung disease with lung sound phenomena present and the ability to sit upright. Exclusion criteria were acute infections, worsening acute heart disease, body mass indexes less than 18 kg/m2 and more than 40 kg/m2, and the presence of pleural effusion. During examination, physicians were unable to see if the patient was wearing a mask (patients' heads were behind an opaque barrier). At the beginning of each examination, a fourth physician randomly instructed patients to put on or take off a three-layer surgical ear loop mask (ear loops were put behind both ears, the mask placed in front of nose and mouth, and aluminum strip nose wire pressed over the nose bridge), and the pulmonologist would auscultate the lungs once and then repeat auscultation after the patient changed their mask status. Subsequently, the fourth physician recorded the pulmonologists' findings.

## Statistical analysis

We evaluated the data from both groups (mask and no-mask) and performed statistical analysis. As a measure of the pulmonologists' agreement in the assessment of lung sounds, we used Fleiss kappa (K).

K assesses the agreement between raters in cases where categorical measures of ordinal or nominal measurement scales were used. Below, we present the results of the analysis of agreement between physicians in the assessment of various respiratory symptoms in two situations—when patients wore a mask and when patients did not wear a mask. Along with the value of K, we also show its 95% confidence interval. If the confidence intervals of the K values in the two situations did not overlap, this means that there was a statistically significant difference in agreement between the two situations. Otherwise, there was no difference.

In addition, for each symptom, the percentage of cases in which all three doctors completely agreed is shown. Software used for statistical analysis was IBM SPSS Statistics 29 (Chicago, IL, USA).

## RESULTS

### Breath sounds

Three pulmonologists evaluated breath sounds in 50 patients in two situations: while the patients were wearing and not wearing masks. They evaluated breath sounds using the following measurement scale: 0 = normal breath sound, 1 = quieter breath sound, 2 = harsh breath sound. The results of the analysis of physician agreement are presented in Table 1.

Physician agreement in the assessment of breath sounds did not differ depending on whether patients wore a mask or not (Table 1). In both situations, physicians completely agreed in 60% of cases. None of the cases resulted in unanimous agreement among the three physicians on the presence of harsh breath sounds.

### Expiration

All three physicians assessed the expiration of 27 patients in two situations (patients with and without a mask). Expiration was assessed using the following scale: 0 = regular, 1 = prolonged. The results of the analysis of physician agreement are shown in Table 1.

Physician agreement on the duration of expiration did not differ depending on whether patients wore mask or not. However, in both situations, the agreement was very low and physicians were in complete agreement only in 26% of cases.

### Abnormal breath sounds

All three physicians assessed the presence of abnormal breath sounds in 28 patients in two situations (patients with and without a mask). They evaluated two sound phenomena: wheezing and crackles. Both phenomena were evaluated using a measuring scale: 0 = sound phenomenon is not present, 1 = sound phenomenon is present. The results of the analysis of physician agreement are shown in Table 1.

Agreement of physicians in the assessment of abnormal breath sounds did not differ based on whether patients wore a mask or not. When assessing the presence of wheezing,

**Table 1  Results of the analysis of the agreement of physicians in the assessment of breath sounds of patients when they wore masks and in cases they did not ($n = 50$).**

| Sound phenomenon assessed | Patients wear a mask | | Patients do not wear a mask | |
|---|---|---|---|---|
| | K (95% CI) | f (%) cases with complete agreement | K (95% CI) | f (%) cases with complete agreement |
| Any breath sound ($n = 50$) | 0.41 (0.27–0.55) | 30 (60%) | 0.40 (0.26–0.54) | 30 (60%) |
| Prolonged expirium ($n = 27$) | −0.09 (−0.31–0.13) | 7 (25.9%) | −0.06 (−0.28–0.16) | 7 (25.9%) |
| Wheezing ($n = 28$) | 0.36 (0.15–0.58) | 18 (64.3%) | 0.41 (0.20–0.62) | 19 (67.9%) |
| Crackles ($n = 28$) | 0.18 (−0.04–0.39) | 11 (39.3%) | 0.19 (−0.03–0.40) | 11 (39.3%) |

Notes. Legend:
K, Fleiss' kappa; 95% CI, 95% confidence interval; f, frequency.

agreement was mild to moderate, and physicians agreed in 64% of cases when patients wore a mask and 68% of cases when patients did not wear a mask. When assessing crackles, agreement was low, and physicians agreed in both situations in 39% of cases.

## DISCUSSION

To our knowledge, no similar study has been published so far. The results of our study showed that there was no significant difference in pulmonologists' agreement in the assessment of breath sounds in all three categories (normal breath sound, duration of expiration, and abnormal breath sound) whether the patient was wearing a mask or not, but there were significant differences among pulmonologists when it came to overall agreement in assessed breath sounds. The difference in auscultation was determined by each pulmonologist because auscultation is a subjective method and interpretations vary widely between physicians (*Xavier et al., 2014*). Auscultation is an essential method in everyday practice that strongly influences future diagnostic and therapeutic workflows. It is performed by various specialists and characterized by its cost effectiveness, availability, simplicity, and transferability. The disadvantages of the mentioned methods are low sensitivity (37%) and acceptable specificity (89%), at least in acute respiratory pathology, all due to high subjectivity and difference in the experience of physicians (*Arts et al., 2020*). Indeed, physicians often differ in their assessments. In published studies, the pulmonary auscultatory skills of pulmonologists were found to be superior to those of medical students, and interns in internal medicine and general practice (*Mangione & Nieman, 1997*). Therefore, in this study only pulmonologists were included. Given the highly subjective nature of this interpretation, inter-listener variability restricts interoperability, with experience varying widely and differing across specialties (*Sarkar et al., 2015*; *Hafke-Dys et al., 2019*). Other sources of heterogeneity may originate from differences in the intrinsic properties of the stethoscope and extrinsic patient-related factors such as obesity, ambient noise, and patient compliance (*e.g.*, crying child). Our study had some limitations. A higher number of patients would lead to more robust conclusions, and the inclusion of physicians from different specialties could confirm our results. It would also be interesting to see whether the type of the mask has an impact on the results. Further studies with more

objective results could be obtained by using digital stethoscopes with recording capabilities to make comparison analysis of the breath sounds captured audio.

## CONCLUSION

The results suggest that wearing surgical face masks during lung auscultation had no impact on agreement in pulmonologist assessment and is therefore an appropriate epidemiological measure in healthcare systems during the pandemic or in any environment with high risk of airborne infection. Wearing masks can enhance the safety of clinicians and health professionals from respiratory infections. Patients should be encouraged to wear masks because our study proved no significant differences in the physician assessment of auscultated breath sounds whether the patients wore a mask or not. Additionally, patients (particularly those who are susceptible) can lower their risk of infection by wearing masks while being certain that surgical mask will not ''mask'' their breath sounds. This was the first study where the influence of a surgical face mask on lung sound examination was assessed, and the results will reassure medical professionals in encouraging patients to wear a surgical face mask knowing it will not change auscultation findings.

### Funding
This study was done with no funding.

### Competing Interests
The authors declare there are no competing interests.

### Author Contributions
- Ivana Folnožić conceived and designed the experiments, performed the experiments, authored or reviewed drafts of the article, and approved the final draft.
- Marija Gomerčić Palčić conceived and designed the experiments, analyzed the data, authored or reviewed drafts of the article, and approved the final draft.
- Matilda Sabljak conceived and designed the experiments, authored or reviewed drafts of the article, and approved the final draft.
- Ena Vučak performed the experiments, prepared figures and/or tables, and approved the final draft.
- Luka Vrbanić conceived and designed the experiments, performed the experiments, prepared figures and/or tables, and approved the final draft.
- Marija Mandić Perić conceived and designed the experiments, performed the experiments, authored or reviewed drafts of the article, and approved the final draft.
- Fanika Mrsić analyzed the data, prepared figures and/or tables, and approved the final draft.
- Aljoša Šikić conceived and designed the experiments, analyzed the data, authored or reviewed drafts of the article, and approved the final draft.

- Ivan Ivanovski conceived and designed the experiments, analyzed the data, authored or reviewed drafts of the article, and approved the final draft.

## Human Ethics

The following information was supplied relating to ethical approvals (i.e., approving body and any reference numbers):

The University Hospital Center Etical board approved to carry out the study within its facilities.

## Ethics

The following information was supplied relating to ethical approvals (i.e., approving body and any reference numbers):

University Hospital Center Sestre Milosrdnice Ethical committee

## Data Availability

The values of different types of lung sounds are available in the Supplemental Files. The patient information has been anonymized.

## Supplemental Information

Supplemental information for this article can be found online at http://dx.doi.org/10.7717/peerj.17368#supplemental-information.

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
