# Peer review of "Wearing surgical face mask has no significant impact on auscultation assessment"

_PeerJ, doi:10.7717/peerj.17368_

## Round 0.1 · original submission · Major Revisions

Thanks for submitting the manuscript. Kindly address the following queries along with the queries raised by the reviewers-

1. The abstract will contain more methodological information.

2. Mention the keywords.

3. In methods, how did you determine the sample size? Which technique was applied to recruit the patients?

4. Add some information about the study setting. Mention the study design.

5. Some inclusion and exclusion criteria for recruiting patients.

6. Mention the statistical software used for data analysis.

**Language Note:** The review process has identified that the English language must be improved. PeerJ can provide language editing services - please contact us at [email protected] for pricing (be sure to provide your manuscript number and title). Alternatively, you should make your own arrangements to improve the language quality and provide details in your response letter. – PeerJ Staff

·

Basic reporting

Overall, the manuscript is well-written in unambiguous language. Sufficient references and background are provided. Figures and tables included are relevant to the hypothesis. However, some tab and column names in the supplemental tables are not in English. Please update them accordingly.

Experimental design

This study is novel and original. The knowledge gap is well-defined. Rigorous investigation was performed. However, the methods were not described in sufficient detail.
- In lines 75-78, it is unclear how many observations were made for each patient by each physician. Based on the supplementary tables, I assume two observations were made for each patient by each physician when the patient was wearing a mask, and was not wearing a mask, respectively. It’s a bit hard to tell from the methods section. It would be great if you could add the relevant information in lines 75-78.
- Only 27 out of 50, and 28 out of 50 patients were assessed for expiration and abnormal breath sounds, respectively. This is a huge attrition rate. Could you provide the reasons why some of the patients were not assessed in the result section? Or could you mention the exclusion criteria in the method section?

Validity of the findings

The low Fleiss kappa of the physician agreement in the assessment of breath sounds within the two patient groups (e.g., patients wear a mask, and patients do not wear a mask) indicates low inter-rater reliability. This study shows that the inter-rater reliability is not significantly different when patients wear a mask vs. when they don’t. However, these findings do now support what the authors are trying to conclude. In lines 120-124, the authors state that this study's results showed no significant difference between pulmonologists' assessment of breath sounds whether the patient was wearing a mask or not. To make such a statement, the authors should evaluate the level of agreement between the observations made when patients wore masks and when patients did not wear masks, instead of the level of agreement between the physicians. Fleiss kappa should be calculated to evaluate the agreement between the assessment made when patients wore masks vs. when the patients did not wear masks.

Additional comments

- It would be great if the authors could provide patient demographics and characteristics since some of the factors could affect the auscultation assessment. How many inpatients and how many outpatients? Patient age group? Any known respiratory pathology history? The reason for getting lung sounds auscultation? The primary reason for their hospital stay or clinic visits? Chest radiography results? Any complications (such as obesity)?
- In line 46, please add the date when you accessed the website. Since the number of deaths is changing over time.
- The current title is not quite informative. Could change it to “Wearing surgical face mask has no significant impact on auscultation assessment.”

·

Basic reporting

Strengths: The manuscript presents a relevant investigation into the impact of mask-wearing on pulmonologist assessments during auscultation. The study design and methodology are well-described, with a clear objective addressing the impact of masks on lung sound assessments.

Areas for Improvement:

The article maintains a clear, professional English language throughout, ensuring readability and comprehension, although certain sections need rephrasing to improve comprehension for an international audience. The introduction sets the context well, providing background information on COVID-19, face mask-wearing, and the importance of lung auscultation. However, further details on patient demographics and the specific type of masks used would enhance clarity and completeness. References to relevant literature are appropriately included, supporting the study's context and findings. The structure of the article adheres to professional standards, with clear sections outlining the methods, results, and discussion. Figures, though not explicitly mentioned in the text, would enhance the comprehensibility of the findings.
The provided ethical approval and Informed Consent lack an English translation, hindering comprehensive evaluation. Translating the Ethical statement as well as the Consent, although not included in the final paper, would aid in ensuring adherence to ethical standards.

Suggestions
- Translate the ethical approval statement and consent into English for proper evaluation and compliance.
- Although already well written and fluent, it's possible to enhance clarity by revising language and improving sentence structure, such as:

Introduction:
"recommended community-wide face mask-wearing" — revise to "recommended community-wide use of face masks."
"There have been several studies showing that face-mask wearing led to reduction in virus transmission" — revise to "Several studies have demonstrated that wearing face masks leads to a reduction in virus transmission."
"helpful in identifying respiratory pathology even when the chest radiography findings are normal" — revise to "helpful in identifying respiratory pathology even when chest radiography findings appear normal."
"advantage is also that it is widely available and inexpensive" — revise to "advantage also lies in its widespread availability and affordability."

Materials & Methods:

"Division of pulmonology, Department of Internal Medicine, Sestre milosrdnice University Hospital Center" — consider rephrasing for clarity: "Division of Pulmonology, Department of Internal Medicine, Sestre milosrdnice University Hospital Center."
"both genders were included" — revise to "both genders were represented."
"then the pulmonologist would auscultate the lungs and then repeat examination after patient has been instructed to put the mask on or take it off" — simplify: "The pulmonologist auscultated the lungs and then repeated the examination after the patient changed their mask status."
"The fourth physician then captured the pulmonologists findings." — rephrase for clarity: "Subsequently, the fourth physician recorded the pulmonologists' findings."

Results:

"did not differ depending on whether patients wore a mask or not" — revise to "did not differ based on whether patients wore a mask."
"It should be noted that in none of the cases did all three physicians agree that patient had harsh breath sound." — rephrase for clarity: "None of the cases resulted in unanimous agreement among the three physicians on the presence of harsh breath sounds."

Discussion:

"As this interpretation is a highly subjective skill, inter-listener variability limits interoperability" — revise to "Given the highly subjective nature of this interpretation, inter-listener variability restricts interoperability."
"where accuracy ranges widely with experience and differs across specialties" — revise to "with experience varying widely and differing across specialties."
"Higher patient number would lead to firmer conclusions" — revise to "A higher number of patients would lead to more robust conclusions."

Conclusions:

"Clinicians and health professionals are safer from respiratory infections when they are wearing masks" — consider rephrasing for clarity: "Wearing masks can enhance the safety of clinicians and health professionals from respiratory infections."
"patients, especially ones susceptible" — revise to "patients, particularly those susceptible."

These suggestions aim to enhance the clarity and flow of the language used in the scientific paper, ensuring a more comprehensible and coherent read for an international audience.

Experimental design

Strengths: The study design demonstrates a rigorous approach, addressing the impact of mask-wearing on lung sound assessments. The blinded examinations conducted by three pulmonologists indicate a well-structured methodology.

Areas for Improvement:

Variability in Instruction: The random instruction for mask usage might introduce variability in assessments. Providing a standardized protocol for mask instructions could minimize this.
Specificity on Mask Types: Lack of information on the specific types of masks used limits understanding. Describing the mask types, fits, or variations would clarify their potential influence on assessments.

Suggestions:

Standardize mask usage instructions to minimize variability in assessments.
Provide detailed descriptions of mask types, fits, or variations to understand their potential impact on assessments.

Validity of the findings

Strengths: The study reveals moderate agreement among pulmonologists in assessing lung sounds regardless of mask usage, indicating consistent evaluations.

Areas for Improvement:

Interpretation of Findings: The manuscript lacks an explicit assessment of the impact and novelty of the findings. Strengthening the conclusions by directly linking them to supported results would enhance validity. However, it encourages meaningful replication by identifying the subjectivity and variability in pulmonologist interpretations during auscultation. While the study's conclusions align with the research question, they would benefit from clearer delineation, directly linking to the study's objectives and supported results.

Transparency in Analysis: Clarify the rationale behind the choice of certain analysis methods to strengthen the validity of results.

Suggestions:
Enhance conclusions by explicitly linking them to supported results and address the impact and novelty of the findings.
Justify the choice of analysis methods used in specific sections to enhance transparency and validity.
The study provides data on breath sounds, expiration duration, and abnormal breath sounds, supporting the conclusions drawn. Besides, certain limitations, such as the small sample size and exclusive inclusion of pulmonologists, are acknowledged. Conclusions are well stated, linked to the original research question.

Additional comments

Ethical Approval Translation and Informed Consent: Address the English translation of the documents to ensure thorough evaluation and compliance (most important issue), although they will not be included in the final publication.
Mask Usage Protocol: Standardize mask usage instructions for consistency in assessments (next most important issue).
Detailed Mask Descriptions: Provide detailed descriptions of mask types used to understand their potential influence on assessments (least important points).

---

## Round 0.2 · Minor Revisions

Thanks for making the necessary changes to the manuscript.

·

Basic reporting

Tables 1-3 can be condensed into a single table, with an extra column showing the auscultation assessment type.

Experimental design

The authors compared the auscultation assessment across different physicians to show that the inter-rater agreement did not get affected by patients wearing or not wearing the mask.

However, to assess if a patient wearing a mask will affect a physician’s auscultation assessment, the authors should also evaluate if the auscultation assessment from the same physician differs before and after the patient wearing a mask. To evaluate that, the authors should report the agreement of the auscultation assessment results from the same physician for patients before and after wearing the mask.

Validity of the findings

The authors currently assessed whether a patient wearing a mask will affect physicians' inter-rater agreement, but failed to assess whether a patient wearing a mask will affect a physician’s auscultation assessment. So the study results that were reported currently do not fully support their study conclusion.

---

## Round 0.3 · accepted · Accept

Thanks for addressing all the comments.

·

Basic reporting

The authors have sufficiently addressed my comments.

Experimental design

The authors have sufficiently addressed my comments.

Validity of the findings

The authors have sufficiently addressed my comments.

·

Basic reporting

The revised paper demonstrates a commendable adherence to professional standards. Its language is clear and technically precise, fostering easy comprehension. The inclusion of thorough literature references provides valuable context, while the structured layout, figures, and shared raw data enhance its credibility. Overall, it stands as a self-contained, coherent contribution to its field, meeting expectations for publication with distinction.

Experimental design

The revised paper aligns impeccably with the standards set forth by the journal. Its research question is well-defined, relevant, and addresses a meaningful knowledge gap within the field. The investigation is conducted rigorously and ethically, adhering to good technical standards. Furthermore, the methods are described in sufficient detail, providing clarity and enabling replication of the study.

Validity of the findings

The revised paper is adhering to the journal's principle of encouraging meaningful replication where rationale and benefits to the literature are clearly articulated. All underlying data provided are robust, statistically sound, and controlled, in accordance with discipline-specific standards. The conclusions are succinctly stated, directly linked to the original research question, and judiciously confined to supporting results, maintaining the paper's integrity.

Additional comments

Given its adherence to the stringent publication criteria outlined by the journal, the revised paper is deemed acceptable for publication. It meets the required standards for clarity, professionalism, methodological rigor, data transparency, and relevance to the field, warranting its inclusion in the journal's esteemed collection of scholarly works.